# Diagnostic Yield of Genetic Testing for Ocular and Oculocutaneous Albinism in a Diverse United States Pediatric Population

**DOI:** 10.3390/genes14010135

**Published:** 2023-01-03

**Authors:** Kyle S. Chan, Brenda L. Bohnsack, Alexander Ing, Andy Drackley, Valerie Castelluccio, Kevin X. Zhang, Hanta Ralay-Ranaivo, Jennifer L. Rossen

**Affiliations:** 1Department of Ophthalmology, Northwestern University Feinberg School of Medicine, Chicago, IL 60611, USA; 2Division of Ophthalmology, Ann & Robert H. Lurie Children’s Hospital of Chicago, Chicago, IL 60611, USA

**Keywords:** ocular albinism, oculocutaneous albinism, diagnostic yield

## Abstract

The diagnostic yield of genetic testing for ocular/oculocutaneous albinism (OA/OCA) in a diverse pediatric population in the United States (U.S.) is unclear. Phenotypes of 53 patients who presented between 2006–2022 with OA/OCA were retrospectively correlated with genetic testing results. Genetic diagnostic yield was defined as detection of pathogenic/likely pathogenic variant(s) matching the anticipated inheritance for that gene–disease relationship. Variant reclassifications of those with variants of uncertain significance (VUS) and without positive diagnostic yield were completed. Overall initial genetic diagnostic yield of OA/OCA was 66%. There was no significant difference (*p* = 0.59) between race and ethnicities (Black (78%), White (59%), Hispanic/Latino (64%)); however, the diagnostic yield of OA (33%) was significantly lower (*p* = 0.007) than OCA (76%). Causative variants in *OCA2* (28%) and *TYR* (20%) were most common. Further, Hermansky–Pudlak syndrome variants were identified in 9% of patients. Re-classification of VUS in non-diagnostic cases resulted in genetic diagnoses for 29% of individuals and increased overall diagnostic yield to 70% of all subjects. There is a high diagnostic yield of genetic testing of patients overall with OA/OCA in a diverse U.S. based pediatric population. Presence or absence of cutaneous involvement of albinism significantly affects genetic diagnostic yield.

## 1. Introduction

Ocular albinism (OA) and oculocutaneous albinism (OCA) encompass a spectrum of inherited disorders that affect an estimated 1 in 18,000 individuals. While OA and OCA present with varying ocular manifestations including iris, retinal pigment epithelium and choroid hypopigmentation, optic nerve abnormalities and foveal hypoplasia, OCA is also associated with cutaneous findings and in some cases systemic syndromes namely Hermansky–Pudlak and Chediak-Higashi syndromes [1].

OCA is usually inherited in an autosomal recessive pattern, whereas, OA is typically x-linked recessive inheritance [2]. Several genes have been linked with specific subtypes of OA and OCA; *TYR* has been associated with OCA type 1, *OCA2* with OCA type 2, *OA1/GPR143* with OA, and various *HPS* genes with Hermansky–Pudlak syndrome [2,3,4,5,6,7]. However, clinical findings may not accurately predict a molecular diagnosis and the spectrum of clinical disease varies widely as many patients with OCA are compound heterozygotes [2,8,9,10,11,12,13].

Several studies in Europe have examined the diagnostic yield of whole genome sequencing or targeted gene panels for albinism with positive yields ranging from 28–91% [8,9,10,11,12,13]. However, the generalizability of these European studies has limited applicability in a diverse United States (U.S.) based population. The purpose of this study was to evaluate the diagnostic yield of genetic testing for a diverse population of pediatric patients in the U.S. clinically suspected of OA/OCA and to identify factors that may affect the diagnostic yield.

## 2. Materials and Methods

A retrospective review identified patients diagnosed with OA or OCA who were evaluated by the Ophthalmology service at the Ann & Robert H. Lurie Children’s Hospital of Chicago between 1 January 2006 and 31 July 2022. This study was approved by the Institutional Review Board of Lurie Children’s Hospital, abided by the tenets of the Declaration of Helsinki, and conducted in accordance with the Health Insurance Portability and Accountability Act. Clinical suspicion for OCA was based on skin and hair hypopigmentation and at least 1 ocular finding such as nystagmus, foveal hypoplasia, fundus hypopigmentation or iris transillumination defects and for OA, absence of cutaneous and hair findings and at least 2 of the above-mentioned ocular findings. Inclusion criteria included prior genetic testing performed for clinical suspicion of OA or OCA. Exclusion criteria were lack of available genetic testing results, absence of ocular manifestations of albinism or the presence of another diagnosis for the ocular manifestations suggestive of OA/OCA.

Data collection included race and ethnicity, family history of albinism, ocular findings, presence of cutaneous involvement (based on clinical observation), history of genetic testing, availability of genetic testing results, genetic variants identified, classification of variant(s), and clinically suspected or, if available, molecular confirmed final diagnosis. Race and ethnicity were recorded from information available in the electronic medical record for the purposes of understanding how race and ethnicity affects genetic testing results. Data included patients who reported as American Indian or Alaskan Native, Asian, non-Hispanic Black or African American (hereafter, Black), Declined or Unknown, Hispanic or Latino (hereafter, Hispanic), Native Hawaiian or Pacific Islander, or non-Hispanic White (hereafter, White). Other was defined as not any of the aforementioned racial and ethnic categories.

Classification of variants identified upon testing prior to the 2015 Richards et al. guidelines for the interpretation of variants, were evaluated and recorded according to a 3-tier system (pathogenic/likely pathogenic (P/LP), variant of uncertain significance (VUS), and benign/likely benign (B/LB)) [14]. After 2015 variants were recorded according to the standardized 5-tier system (P, LP, VUS, B, LB) [14]. Positive genetic diagnostic yield was defined as detection of P/LP variant(s) matching the anticipated inheritance for that gene–disease relationship (autosomal recessive for *TYR*, *OCA2*, *TYRP1*, *SLC45A2*, *SLC24A5*, *C10orf11*, *HSP* genes, *LYST*, and *DCT* and x-linked recessive for *GPR143*) [14,15].

Demographic and clinical information, as well as genetic testing results, were compiled using descriptive statistics. Comparison of diagnostic yield was performed within variables of interest using Chi-square analysis. A *p*-value of 0.05 was deemed statistically significant for all analysis.

Ten unique previously identified VUS from seven patients with non-diagnostic genetic testing were re-interpreted and classified between September and October 2022. All 7 of the patients had ocular findings of albinism and 4/7 also had cutaneous findings. These re-interpretations included a review of publicly available databases (e.g., ClinVar [16], Leiden Open Variation Database [17]) and the latest biomedical literature for data relevant to each variant. Minor allele frequency and gene constraint data were obtained from the Genome Aggregation Database [18] v2.1.1 and v3.1.2. In silico pathogenicity predictions were performed using REVEL [19] (benign: ≤0.290; pathogenic: ≥0.644 [20] and the splicing prediction algorithms SpliceSiteFinder-like [21], MaxEntScan [22], NNSplice [23], and GeneSplicer [24] (predicted impact: ∆ ≥ 10% between wild-type and variant), and SpliceAI [25] (predicted impact: ≥0.2; no impact: <0.2). Classifications were derived in accordance with the recommendations established by the 2015 American College of Medical Genetics and Genomics—Association for Molecular Pathology (ACMG-AMP) technical standards for sequence variant interpretation [14]. Updated variant classifications were then compared to the original classifications.

To investigate crucial signaling pathways in our OA/OCA cohort, selection of unranked candidate genes was guided by detected variants in our patients that underwent commercial genetic testing and submitted for pathway enrichment analysis. Annotations were carried out with the gProfiler toolset, which utilizes Gene Ontology (GO), Kyoto Encyclopedia of Genes and Genomes, Reactome, Human Protein Atlas, Comprehensive Resource of Mammalian Protein Complexes, and Human Phenotype Ontology databases. Enrichment was tested using the native g:SCS significance algorithm, corresponding to a query-wide false discovery rate corrected threshold of α = 0.05 [26].

## 3. Results

One-hundred and eighteen patients with a clinical diagnosis of OA or OCA were identified, of which fifty-three (45%) had previously undergone genetic testing (Appendix A). In the cohort with genetic testing results, the average age at initial evaluation was 1.3 ± 2.0 years (median 0.6, range 0.05–10) and two-thirds were male (Table 1). Forty-two percent of the patients identified as White while 26% were Hispanic and 17% were Black (Figure 1). In total, 17 of 48 (35%) patients endorsed a family history of albinism, while information regarding family history of albinism for 5 patients were unavailable. Forty-one (77%) patients had a diagnosis of OCA and 12 (23%) had a diagnosis of OA, based on clinical findings (absence or presence of cutaneous findings). Five (9%) patients were diagnosed with Hermansky–Pudlak syndrome based on genetic testing. Only one patient (Patient #6 in Appendix A) with Hermansky-Pudlak syndrome had only ocular findings without cutaneous involvement.

On ophthalmic examination, 47 (89%) patients had nystagmus and 45 (85%) had documented foveal hypoplasia. Fewer patients had fundus hypopigmentation (36 patients, 68%) or iris transillumination defects (20 patients, 38%). 

The overall positive diagnostic yield of genetic testing in the combined OA and OCA cohort was 66% (35 of 53 patients, Table 2) before variant reclassification. There was no significant difference in diagnostic yield by race or ethnicity (*p* = 0.59). However, cutaneous involvement correlated with a statistically significant higher yield (76% vs. 33%, *p* = 0.007). There was no significant difference in diagnostic yield (*p* = 0.82) when comparing the four following ocular manifestations of OCA/OA: nystagmus (70%), fundus hypopigmentation (69%), foveal hypoplasia (69%), and iris transillumination defects (80%). For the 13 patients with all 4 ocular features, 92% had initial positive diagnostic yield, which increased to 100% after variant reclassification.

The most common genes with P or LP variants were *OCA2* (28%) and *TYR* (20%) (Table 3). Other genes with P or LP variants included *HPS5* (6%), *TYRP1* (4%), *HPS1* (2%), *HPS6* (2%), *SLC45A2* (2%), and *OA1*/*GPR143* (2%). 

Of the ten VUS re-interpreted in this study, two received upgraded classifications, yielding two additional genetic diagnoses. Re-classification of VUS in non-diagnostic cases resulted in genetic diagnoses for 29% of individuals that were re-interpreted and increased the overall diagnostic yield of study patients to 70% (Table 2). For patient 16 (Appendix A), the variant NM_000273.3(GPR143):c.455+3A>G was classified as pathogenic in this study, supporting this patient’s diagnosis of OA (ACMG-AMP criteria applied: PS3, PS4_moderate, PP4). This classification discordance appears to be due to differences in application of ACMG-AMP evidence criteria rather than availability of new data. For patient 28, the variant NM_000372.5(TYR):c.649C>G (p.Arg217Gly) was classified as LP (ACMG-AMP criteria applied: PM2, PM3, PM5, PS4_supporting). This variant was present with a pathogenic start-loss variant in TYR and thus was consistent with a diagnosis of OCA. 

Pathway analysis of 33 distinct variant genes (AP3D1, ARHGEF18, CACNA2D4, CDH23, CNGA3, COL18A1, CTNNA1, CYP4V2, FSCN2, GPR143, GPR179, HMX1, HPS1, HPS4, HPS5, HPS6, IDH3A, KIF7, LRP2, LYST, MC1R, MPDZ, MTTP, OCA2, PCDH15, RAB27A, RIMS1, RP1, SLC45A2, SLC7A14, TYR, TYRP1, ZNF423) from our 53 patients using gProfiler showed that molecular function, biological process, and cellular component annotations all featured enriched terms consistent with pigmentation at the level of the melanosome in our gene set (Figure 2). Significant terms that shared overlap with genes identified as having positive diagnostic yield were melanin biosynthetic process (OCA2, TYR, TYRP1; GO:0042438; *p* = 4.16 × 10^−7^), pigment biosynthetic process (GPR143, SLC45A2; GO:0046148; *p* = 5.75 × 10^−7^), melanocyte differentiation (HPS4, HPS6; GO:0030318; *p* = 8.11 × 10^−7^), and BLOC complex (HPS1, HPS5; GO:0031082; *p* = 7.29 × 10^−6^).

## 4. Discussion

Both OA and OCA are commonly diagnosed by ophthalmologists due to infantile-onset of nystagmus and decreased visual function. While clinical diagnosis may be apparent by iris and fundus hypopigmentation, genetic confirmation is useful for understanding visual prognosis and in some cases identifying associated systemic diseases. The current study aims to address the limited information regarding diagnostic yield in OA and OCA in the diverse U.S. population. 

A positive diagnostic yield of 70% in this U.S.-based population of patients with OCA and OA, after consideration of variant reclassification, falls within the range from prior studies in other countries with rates ranging from 28% to 91%. None of these prior studies were completed in the U.S., with the majority of reports coming from the United Kingdom, a country that remains ahead of the U.S. in many population-based ocular genetic studies due to accessibility of covered testing with universal healthcare. In the United Kingdom, Lenassi et al. reported a 91% positive diagnostic yield in 32 pediatric patients with albinism and low vision using 3 targeted panels [10]. Jackson et al. reported a 56% diagnostic yield in 9 pediatric patients with either whole genome sequencing or a targeted panel, but only a 28% positive diagnostic yield in 114 patients who underwent whole genome sequencing as part of a government genomic initiative (100,000 Genomes Project) [11]. It is possible that the yield in the government genomic initiative is low due to the exclusion of patients with previously identified molecular diagnoses from targeted panels. In a predominant French cohort, Lasseaux et al. reported a 72% diagnostic yield using a targeted next-generation sequencing panel of 19 genes in a cohort of 990 patients with albinism [8]. Hovnik et al. reported a 80% diagnostic yield in 25 Slovenian pediatric patients tested with a targeted panel, and observed that phenotype did not predict genotype, although no formal statistical analysis was performed [13].

In the current study, genetic testing of OCA compared to OA had a higher success in confirming the clinical diagnosis. While this may be due to the higher number of patients with both ocular and cutaneous findings in our cohort (85%), this distribution reflects the proportion of OA to OCA (approximately 10% of albinism cases) in the general population. Importantly, cutaneous involvement, was a strong indicator for positive diagnostic yield and is likely related to the prevalence of the most common genes associated with OCA, *TYR* and *OCA2.* Similarly, Chan et al. showed a high positive diagnostic yield in *TYR* and *OCA2* in families with albinism and cutaneous involvement [12]. The lower rate of positive diagnostic yield in patients without cutaneous involvement is likely secondary to the lower prevalence as well as the wide spectrum of albinism phenotypes. While straightforward Mendelian genetics accounts for the majority of identified gene variants in OCA, there is increasing information regarding compound heterozygote inheritance patterns that are associated with more subtle clinical findings. For example, in our reclassification of ten VUS, eight did not significantly change classifications upon re-interpretation due to insufficient new data. However, multiple of these variants are complex in their apparent contribution to disease, such as conferring an increased risk for OCA in a non-Mendelian manner or acting as a component of a disease-associated haplotype, requiring at least one additional variant to be clinically significant. Because of these variants’ complex and not yet clearly established association with disease, laboratories have discrepant practices for their classifications, leading to discordant interpretations. 

Our rates of ocular manifestations: 89% with nystagmus, 85% with foveal hypoplasia, 68% with fundus hypopigmentation, and 38% with iris transillumination defects were close or slightly lower to the reported rates in the two papers that mentioned their findings. The previously reported rates of ocular features were: 86.4–100% with nystagmus, 75.0–92% with foveal hypoplasia, 72.7% with fundus hypopigmentation and 50.0–58% with iris transillumination defects [9,12].

Additionally, some patients with nystagmus, foveal hypoplasia, fundus hypopigmentation, and iris transillumination defects may have a non-albinotic condition, and therefore lower rate of molecular diagnosis with targeted panel testing for albinism [2,12]. Other conditions such as Aland Island Eye disease caused by a mutation in *CACNA1F* and Waardenburg Syndrome associated with *PAX3* mutations can have similar clinical findings of albinism on ocular exam [2]. Additionally, patients with recessive mutations in *SLC38A8* can have foveal hypoplasia and visual pathway misrouting, but this is a genetically distinct entity from OA/OCA [27]. It is also possible that there are novel genes and variants yet to be discovered that cause ocular albinism, which supports expanded testing in this patient population.

Pathway analyses from variant genes defined a priori from our patient population suggest that pigmentation and melanin biosynthetic processes are strongly implicated. Genes syndromic for conditions featuring albinism phenotypes distinct from OA and OCA also shared considerable overlap with our enriched GO terms, including BLOC, BLOC-2 and BLOC-3 complexes sharing associations with Hermansky–Pudlak syndrome, and melanosome and pigment granule organization sharing associations with Chediak-Higashi syndrome. However, the fact that many subtypes of OA and OCA still lack genetic classification suggests either that certain phenotypes represent a spectrum of pigmentation defects, or that we have not fully captured the diversity of OA and OCA genotypes.

Limitations of the study include the retrospective and single-center nature of the study, which renders the results subject to bias. The sample size of the study limits the statistical power of the conclusions. Cutaneous involvement was determined based on clinical observation alone without confirmatory tests, such as hair bulb incubation test, which could provide more certainty to the diagnosis. Additionally, ocular findings were determined in the office setting, which can be difficult in a young patient population, particularly for those with nystagmus. Even identification of foveal hypoplasia on imaging in older children who are able to cooperate with the exam can still be technically challenging. No patients underwent visual evoked potentials, but two patients did have normal electroretinograms. There was no consistent genetic panel selected for patients or date of testing, as the specific test was ordered per provider preference over several years, which may limit the conclusions regarding the diagnostic yield of genetic testing. However, this does depict a real-world diagnostic yield over time and is consistent with publications in the literature which frequently reported multiple different testing modalities in their studies [8,9,10,11,12,13]. Additionally variant reclassification of patients without a molecular diagnosis on initial testing, had reclassification of their variants, which helps adjust for the year of testing variable. Future studies will seek to investigate diagnostic yield of prospective genetic testing in a multi-center setting and with expanded testing to identify new genes and more rare variants, particularly in the under-studied condition OA.

## 5. Conclusions

Molecular genetic testing for the evaluation of OA and OCA has high diagnostic yield in a diverse U.S. based population with variant reclassifications improving yield even further. Cutaneous involvement significantly affected diagnostic yield, while race and ethnicity had no effect. Patients with syndromes associated with systemic illnesses were identified in 9% of this cohort by the results of the genetic testing, all with Hermansky–Pudlak Syndrome. This study affirms the utility of genetic testing to guide the management of patients and families with OA and OCA.

## Figures and Tables

**Figure 1 genes-14-00135-f001:**
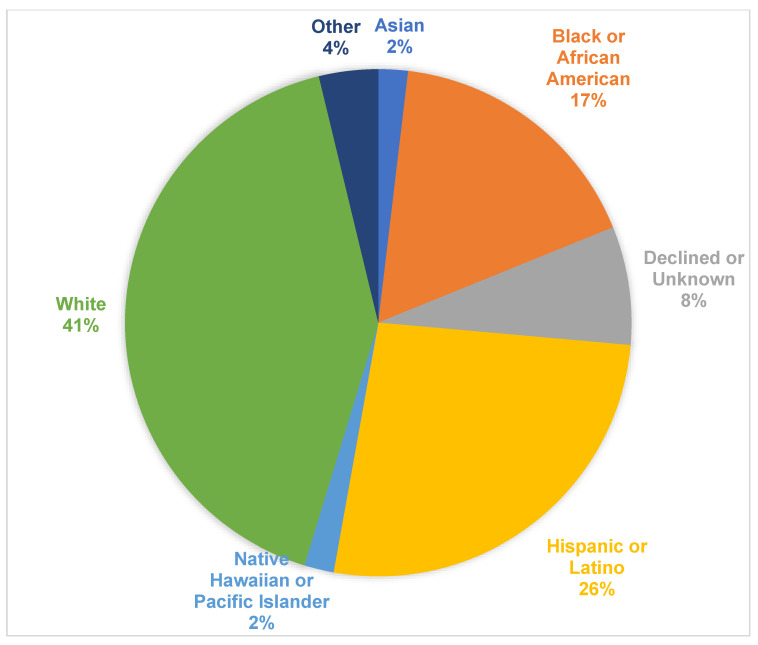
Race/Ethnicity.

**Figure 2 genes-14-00135-f002:**
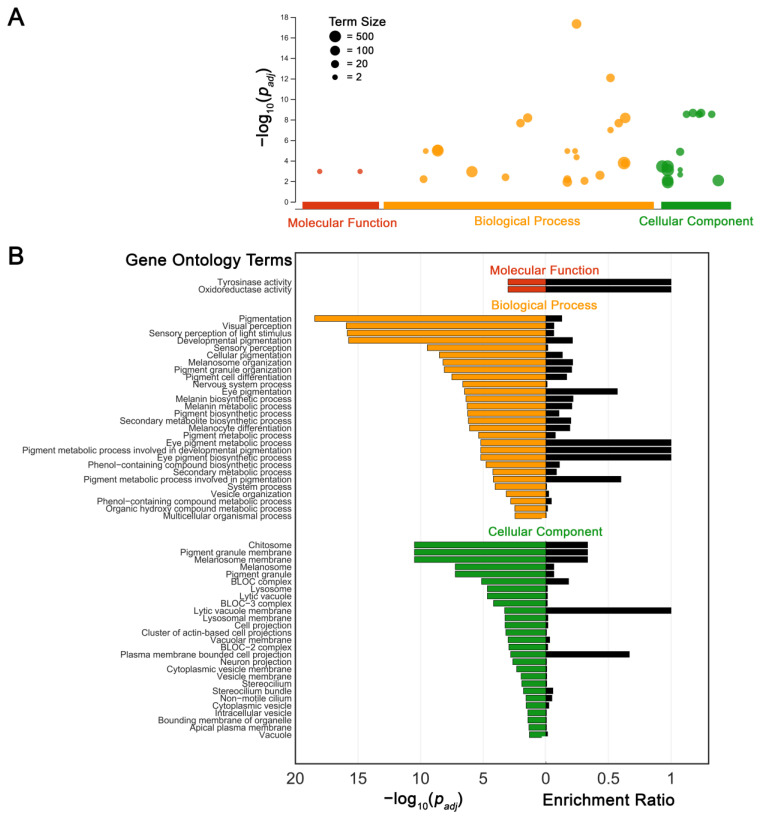
Pathway enrichment analysis depicting significantly enriched gene ontology (GO) terms belonging to (**A**) categories: biological process, molecular function, and cellular component, with corresponding false-discovery rate (FDR) corrected significance values. Term size corresponds to total number of genes associated with a given GO term. (**B**) Detailed list of GO terms with their corresponding enrichment ratio expressed as the number of overlapping genes divided by total term size.

**Table 1 genes-14-00135-t001:** Demographic and clinical characteristics of the 53 patients evaluated for OA/OCA with reviewable genetic testing results.

Characteristic	N (%)
Age (years ± standard deviation)	1.3 ± 2.0
**Sex**	
Male	35 (66)
Female	18 (34)
Family history of suspected albinism	17/48 (35)
**Clinical diagnosis**	
Oculocutaneous albinism	41 (77)
Ocular albinism	12 (23)
**Ophthalmic exam findings**	
Nystagmus	47 (89)
Fundus hypopigmentation	36 (68)
Foveal hypoplasia	45 (85)
Iris transillumination defects	20 (38)

**Table 2 genes-14-00135-t002:** Number of patients with positive and negative diagnostic yield based on presence of patient factors. Comparison within variables performed via Chi-square analysis.

	Number of Patients with Positive Diagnostic Yield (%)	Number of Patients with Negative Diagnostic Yield (%)	*p*-Value
Overall After variant reclassification	35 (66) 37 (70)	18 (34) 16 (30)	--
**Cutaneous involvement**			
Yes No	31 (76) 4 (33)	10 (24) 8 (67)	**0.007 ***
**Ocular Manifestations**			
Nystagmus Fundus hypopigmentation Foveal hypoplasia Iris transillumination defects	33 (70) 25 (69) 31 (69) 16 (80)	14 (30) 11 (31) 14 (31) 4 (20)	0.82
**Race and ethnicity**			
Asian Black or African American Declined or Unknown Hispanic or Latino Native Hawaiian or Pacific Islander Other White	0 (0) 7 (78) 3 (75) 9 (64) 1 (100) 2 (100) 13 (59)	1 (100) 2 (22) 1 (25) 5 (36) 0 (0) 0 (0) 9 (41)	0.59

* is statistically significant result.

**Table 3 genes-14-00135-t003:** Percentage of patients who have P/LP variants in genes associated with positive diagnostic yield.

Gene	N (%)
*OCA2* (%)	15 (28)
*TYR* (%)	11 (20)
*HPS5* (%)	3 (6)
*TYRP1* (%)	2 (4)
*HPS1* (%)	1 (2)
*HPS6* (%)	1 (2)
*SLC45A2* (%)	1 (2)
*OA1/GPR143* (%)	1 (2)
% Taken from total number of patients in the study (53)	

## Data Availability

Not applicable.

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
