# Peer review of "Diagnostic Yield of Genetic Testing for Ocular and Oculocutaneous Albinism in a Diverse United States Pediatric Population"

_genes, 2023, doi:10.3390/genes14010135_

Round 1
Reviewer 1 Report
I would like to thank the authors for their excellent manuscript entitled Diagnostic Yield of Genetic Testing for Ocular and Oculocutaneous Albinism in a Diverse United States Pediatric Population.
This manuscript by Chan et al reviews a cohort of patients from the pediatric population in the United States. Fifty three of 118 patients had a genetically confirmed diagnosis of OA or OCA. The manuscript describes the genetic diagnostic yield of OCA and OA and concludes that cutaneous involvement is a strong indicator in these patients. The reclassification of VUS improved this diagnostic yield. Surprisingly, the most common P or LP variant was OCA2, which has the highest prevalence in African populations, yet there was no significant difference between race or ethnicity and the largest race/ethnicity patients are classified as White and Hispanic/Latino.
This is an elegant, well written and detailed retrospective study that contributes greatly to the interpretation and clinical significance of genetic testing in OA and OCA. This is the first study to document the genetic diagnostic yield in pediatric patients from the United States and highlights the need for targeted genetic testing in patients suspected of OCA or OA, with or without cutaneous involvement.
I would appreciate it if the authors could address the points listed below.
Points to consider:
Page 2, line 62: How was cutaneous involvement defined and was it based on observation or implicated by another method e.g. hair bulb incubation test?
Page 2, line 85: Did the 10 patients with previously identified VUS have any ophthalmic or cutaneous signs of albinism?
Page 3, line 106: Replace ‘One-hundred eighteen patients..’ with ‘One-hundred and eighteen patients…’
Page 3, line 110 (Table 1): The race and ethnicity data may be better represented graphically e.g. a pie chart.
Page 3, line 116: Patient #4 in Supplemental Table 1 is listed as ‘Yes’ under Cutaneous findings. Did the authors mean Patient #5?
Page 3, line 117: The mean age of the patients on presentation was 1.3±2.0 years and most patients presented with nystagmus (89%) and foveal hypoplasia (85%). It is often challenging to examine the fundus of patients of this age, particularly with nystagmus. How was this performed? Were any of the patients assessed under sedation?
Page 3, line 119: As ophthalmic examination can be challenging in this cohort of patients several pediatric centres, including those quoted (Lenassi et al, 2020; Jackson et al., 2020; Chan et al., 2021; Hovnik et al., 2021) utilize electrophysiological tests, such as the visual evoked potential in the diagnosis of OCA and OA. Did any of the patients in this cohort undergo electrophysiological testing? If so, can the authors include the outcome of these tests in Table 1?
Figure 3, Table 1: How does the incidence of foveal hypoplasia and nystagmus in this pediatric population compare with the other studies quoted?
Page 4, line 125: Replace significant different in diagnostic..’ with ‘significant difference in diagnostic..’
Page 4, line 123: The authors compare positive diagnostic yield with cutaneous involvement, and with patient race and ethnicity. Did they compare positive diagnostic yield with any of the ocular features? This would be helpful for prognostic purposes and provide indicators for genetic testing other than cutaneous involvement.
Page 6, line 208: The authors highlight patients with non-albinotic disorders and ocular features similar to albinism. Did any of the 53/118 patients identified on genetic testing have a mutation in SLC38A8 (Poulter et al. Recessive mutations in SLC38A8 cause foveal hypoplasia and optic nerve misrouting without albinism. Am. J. Hum. Genet. 93: 1143-1150, 2013)? As these individuals present with foveal hypoplasia and visual pathway misrouting and are often misdiagnosed with OCA or OA, the authors may mention in the discussion that SLC38A8 is a genetically distinct disorder from OCA and OA.
Author Response
We appreciate the time and effort this reviewer has given our submission. We have responded to each comment below.
Point 1: Page 2, line 62: How was cutaneous involvement defined and was it based on observation or implicated by another method e.g. hair bulb incubation test?
Response 1: Cutaneous involvement was based on observation of pale skin and hair. We did not do a hair bulb incubation test, but that is an excellent point by the reviewer and we have amended our Materials and Methods (Page 2, Line 63) limitations section to reflect this (Page 8, lines 281-283).
Point 2: Page 2, line 85: Did the 10 patients with previously identified VUS have any ophthalmic or cutaneous signs of albinism?
Response 2: All of the patients had ophthalmic signs and 4 also had cutaneous signs of albinism. This was added to the paper on Page 2, lines 86-87).
Point 3: Page 3, line 106: Replace ‘One-hundred eighteen patients..’ with ‘One-hundred and eighteen patients…’
Response 3: Thank you for your edit. This has been corrected (Page 3, line 124).
Point 4: Page 3, line 110 (Table 1): The race and ethnicity data may be better represented graphically e.g. a pie chart.
Response 4: We have changed the race and ethnicity data into a pie chart. Thank you for this excellent comment. This is now Figure 1 on Page 4.
Point 5: Page 3, line 116: Patient #4 in Supplemental Table 1 is listed as ‘Yes’ under Cutaneous findings. Did the authors mean Patient #5?
Response 5: Thank you for identifying this error. The correct patient is Patient #6 (has Hermansky-Pudlak syndrome with ocular but no cutaneous findings) and the manuscript was edited (Page 3, line 134).
Point 6: Page 3, line 117: The mean age of the patients on presentation was 1.3±2.0 years and most patients presented with nystagmus (89%) and foveal hypoplasia (85%). It is often challenging to examine the fundus of patients of this age, particularly with nystagmus. How was this performed? Were any of the patients assessed under sedation?
Response 6: Clinical information was obtained primarily at clinic visits. When patients got older, OCT was attempted but often limited due the nystagmus. This is an excellent point to consider and an addition to the limitations section was added on Page 8, line 283-287.
Point 7: Page 3, line 119: As ophthalmic examination can be challenging in this cohort of patients several pediatric centres, including those quoted (Lenassi et al, 2020; Jackson et al., 2020; Chan et al., 2021; Hovnik et al., 2021) utilize electrophysiological tests, such as the visual evoked potential in the diagnosis of OCA and OA. Did any of the patients in this cohort undergo electrophysiological testing? If so, can the authors include the outcome of these tests in Table 1?
Response 7: On further review of the charts, no patients underwent visual evoked potential. We did have two patients with ERGs to rule out retinal dystrophies as the cause of nystagmus (both were normal) which was added to the limitations section on Page 8, line 287-288.
Point 8: Figure 3, Table 1: How does the incidence of foveal hypoplasia and nystagmus in this pediatric population compare with the other studies quoted?
Response 8: Thank you for this thoughtful comment. We have addressed it in our manuscript as follows on Page 7, lines 251-256: “Our rates of ocular manifestations: 89% with nystagmus, 85% with foveal hypoplasia, 68% with fundus hypopigmentation, and 38% with iris transillumination defects were close or slightly lower to the reported rates in the two papers that mentioned their findings. The previously reported rates of ocular features were: 86.4-100% with nystagmus, 75.0-92% with foveal hypoplasia, 72.7% with fundus hypopigmentation and 50.0-58% with iris transillumination defects [9,12].”
Point 9: Page 4, line 125: Replace significant different in diagnostic..’ with ‘significant difference in diagnostic..’
Response 9: Thank you for your edit. This has been corrected on Page 4, line 157.
Point 10: Page 4, line 123: The authors compare positive diagnostic yield with cutaneous involvement, and with patient race and ethnicity. Did they compare positive diagnostic yield with any of the ocular features? This would be helpful for prognostic purposes and provide indicators for genetic testing other than cutaneous involvement.
Response 10: This is an excellent suggestion. This information has been included in the manuscript on Page 4, lines 159-163 and Table 2. The manuscript additions include: “There was no significant difference in diagnostic yield (p=0.82) when comparing the 4 following ocular manifestations of OCA/OA: nystagmus (70%), fundus hypopigmentation (69%), foveal hypoplasia (69%), and iris transillumination defects (80%). For the 13 patients with all 4 ocular features, 92% had initial positive diagnostic yield, which increased to 100% after variant reclassification.”
Point 11: Page 6, line 208: The authors highlight patients with non-albinotic disorders and ocular features similar to albinism. Did any of the 53/118 patients identified on genetic testing have a mutation in SLC38A8 (Poulter et al. Recessive mutations in SLC38A8 cause foveal hypoplasia and optic nerve misrouting without albinism. Am. J. Hum. Genet. 93: 1143-1150, 2013)? As these individuals present with foveal hypoplasia and visual pathway misrouting and are often misdiagnosed with OCA or OA, the authors may mention in the discussion that SLC38A8 is a genetically distinct disorder from OCA and OA.
Response 11: None of our patients had a P, LP or VUS in SLC38A8. You make an excellent point to include this reference which has since been added to our discussion section on Page 7 lines 262-264.
Reviewer 2 Report
Reviewer’s Comments
1. Line 19; make the genetic variants italic TYR and OCA2.
2. Line 78, 79: authors discussed all the loci and causative genes of autosomal recessive genes except DCT harbouring locus 8 of OCA. It is suggested to include the variants of OCA8 for diagnostic yield or discuss the reason otherwise.
3. Lines 89-92, in silico tools applied for pathogenicity or likely pathogenic effect, mention the cut-off values or the scores for each test.
4. Line 127-128, Table 2, authors performed the association of different variables including ethnic or racial groups and cutaneous phenotype. It is also suggested to compare the nystagmus, trans illumination and visual acuity.
5. Line 152-155, please explain why the OCA genes are not included for pathway analysis.

Author Response
Point 1: Line 19; make the genetic variants italic TYR and OCA2.
Response 1: Thank you for your edit. This has been corrected on Line 19.
Point 2: Line 78, 79: authors discussed all the loci and causative genes of autosomal recessive genes except DCT harbouring locus 8 of OCA. It is suggested to include the variants of OCA8 for diagnostic yield or discuss the reason otherwise.
Response 2: Thank you for your edit. DCT has been added to the list of genes on Page 2, line 80 and a new reference was included.
Point 3: Lines 89-92, in silico tools applied for pathogenicity or likely pathogenic effect, mention the cut-off values or the scores for each test.
Response 3: Thank you for this suggestion, we have edited the manuscript on Page 2, lines 92-95. In silico pathogenicity predictions were performed using REVEL[19] (benign: ≤ 0.290; pathogenic: ≥ 0.644[20,new reference Pejaver et al 2022 https://pubmed.ncbi.nlm.nih.gov/36413997/] and the splicing prediction algorithms SpliceSiteFinder-like [21], MaxEntScan[22], NNSplice[23], and GeneSplicer[24] (predicted impact: ∆ ≥ 10% between wild-type and variant), and SpliceAI[25] (predicted impact: ≥ 0.2; no impact: < 0.2).
Point 4: Line 127-128, Table 2, authors performed the association of different variables including ethnic or racial groups and cutaneous phenotype. It is also suggested to compare the nystagmus, trans illumination and visual acuity.
Response 4: This is an excellent suggestion. This information has been included in the manuscript on Page 4, lines 159-163 and Table 2. The manuscript additions include: “There was no significant difference in diagnostic yield (p=0.82) when comparing the 4 following ocular manifestations of OCA/OA nystagmus (70%), fundus hypopigmentation (69%), foveal hypoplasia (69%), and iris transillumination defects (80%). For the 13 patients with all 4 ocular features, 92% had initial positive diagnostic yield, which increased to 100% after variant reclassification.”
We did not include visual acuity as the reliability of testing was low in this pediatric population and varied by patient at different ages.
Point 5: Line 152-155, please explain why the OCA genes are not included for pathway analysis.
Response 5: Thank you for your comments. We included all syndromic genes implicated in all described OA and OCA genotypes in our pathway analysis, (i.e. TYR for OCA1, TYRP1 for OCA3, SLC45A2 for OCA 4 and 6, etc.). The gProfiler input for our pathway analysis featured a list of 33 genes, chosen on the basis of having any identified variants from our 53 patients that have undergone genetic testing. We recognize from your feedback that this may not be clear in the manuscript and have made changes in the methods and results section describing our pathway analysis to improve readability on Page 3, line 116 and on Page 5, lines 186-189.